# Pathways of Angiogenic and Inflammatory Cytokines in Multiple Myeloma: Role in Plasma Cell Clonal Expansion and Drug Resistance

**DOI:** 10.3390/jcm11216491

**Published:** 2022-11-01

**Authors:** Assunta Melaccio, Antonia Reale, Ilaria Saltarella, Vanessa Desantis, Aurelia Lamanuzzi, Sebastiano Cicco, Maria Antonia Frassanito, Angelo Vacca, Roberto Ria

**Affiliations:** 1Department of Biomedical Sciences and Human Oncology, Section of Internal Medicine “G. Baccelli”, University of Bari Medical School, 70124 Bari, Italy; 2Myeloma Research Group, Australian Centre for Blood Diseases, Central Clinical School, Monash University—Alfred Health, Melbourne 3004, Australia; 3Department of Biomedical Sciences and Human Oncology, Pharmacology Section, University of Bari Aldo Moro Medical School, 70124 Bari, Italy; 4General Pathology Unit, Department of Biomedical Sciences and Human Oncology, University of Bari Medical School, 70124 Bari, Italy

**Keywords:** multiple myeloma, angiogenic cytokines, angiogenic mimicry, osteoblastic niche, vascular niche

## Abstract

Multiple myeloma (MM) is the second most common hematological malignancy, and despite the introduction of innovative therapies, remains an incurable disease. Identifying early and minimally or non-invasive biomarkers for predicting clinical outcomes and therapeutic responses is an active field of investigation. Malignant plasma cells (PCs) reside in the bone marrow (BM) microenvironment (BMME) which comprises cells (e.g., tumour, immune, stromal cells), components of the extracellular matrix (ECM) and vesicular and non-vesicular (soluble) molecules, all factors that support PCs’ survival and proliferation. The interaction between PCs and BM stromal cells (BMSCs), a hallmark of MM progression, is based not only on intercellular interactions but also on autocrine and paracrine circuits mediated by soluble or vesicular components. In fact, PCs and BMSCs secrete various cytokines, including angiogenic cytokines, essential for the formation of specialized niches called “osteoblastic and vascular niches”, thus supporting neovascularization and bone disease, vital processes that modulate the pathophysiological PCs–BMME interactions, and ultimately promoting disease progression. Here, we aim to discuss the roles of cytokines and growth factors in pathogenetic pathways in MM and as prognostic and predictive biomarkers. We also discuss the potential of targeted drugs that simultaneously block PCs’ proliferation and survival, PCs–BMSCs interactions and BMSCs activity, which may represent the future goal of MM therapy.

## 1. Introduction

The balance between different cytokines is essential for tumor growth and progression. The proportion of inflammatory infiltrates and angiogenesis is determined by the vascular response generated by these pro-inflammatory factors [1]. This also occurs in multiple myeloma (MM), an incurable plasma cell (PC) dyscrasia that accounts for approximately 10% of hematological malignancies [2]. Typically, MM evolves from a pre-malignant stage termed a monoclonal gammopathy of undetermined significance (MGUS), which is observed in 1% of adults over 25 years of age. Every year, 1–2% of MGUSs progress to the malignant stage (active) of MM. A small subset of patients has an intermediate clinical phenotype between MGUS and MM, asymptomatic or smoldering multiple myeloma (SMM).

MM is a prototypic disease in which PCs subvert the local bone marrow microenvironment (BMME) to support their growth, elude immune surveillance and develop resistance to chemotherapy and/or immunotherapy. Hematopoietic stem cells (HSCs), mesenchymal stem cells (MSCs), bone marrow stromal cells (BMSCs), including fibroblasts (FBs), osteoblasts, osteoclasts, chondrocytes, endothelial cells (ECs) and endothelial progenitor cells (EPCs), coexist within the BMME with clonal malignant PCs, B and T lymphocytes, neutrophils, macrophages, mast cells, NK cells, erythrocytes, megakaryocytes, platelets and the extracellular matrix (ECM) [2]. The non-cellular compartment, on the other hand, includes soluble growth factors (i.e., interleukin (IL)-6, insulin-like growth factor (IGF)-1, vascular endothelial growth factor (VEGF), fibroblast growth factor (FGF)-2, B-cell activating factor (BAFF), a proliferation-inducing ligand (APRIL), and stromal cell-derived factor (SDF-1)), adhesion molecules and extracellular vesicles [3,4].

Notably, the interaction between malignant PCs and BMSCs in the BMME is essential for the evolution of MM, for example through the production of IL-6 and the priming of the nuclear factor kappa-light-chain-enhancer of activated B cell (NF-κB) signaling [5].

The lack of an accurate prognostic biomarker panel in MM has led to recent research investigating angiogenic inflammatory cytokines and growth factors released in the BMME.

The identification of these “biomolecules” takes place in a unique site of the tumor BMME named the “vascular niche”, in which PCs survive undisturbed and proliferate [6]. The vascular niche originates from the de novo production of a vascular tree via the processes of neoangiogenesis, vasculogenesis and vascular mimicry [7]. The vascular niche allows PCs to leave the osteoblastic niche close to the endosteum and to enter the vascular system by migrating through the endothelium [5].

The deregulated conditions caused by hypoxic, proinflammatory and immuno-evasive conditions that occur in the “vascular niche” are currently under rigorous investigation to better understand the underlying mechanisms of cancer cells’ outgrowth and drug resistance in MM.

Cytokines play a critical role in PCs’ expansion, MM progression, cell-to-cell adhesion, angiogenesis and vasculogenesis. In general, the prevalence of proinflammatory cytokine secretion [IL-1, IL-6, IL-12, IL-15, IL-16, IL-17, IL-18, IL-22, IL-23, tumor necrosis factor (TNF)-α and interferon (IFN)-γ)] versus anti-inflammatory secretion [heat shock proteins (HSPs), IL-1 receptor (IL1R)α, IL-4, IL-10, IL-11, transforming growth factor (TGF)-β1, and lipoxin A4)] in MM patients is contingent on genetic alterations that originate from the cells within the microenvironment [8]. Consequently, MM ECs have an “activated” phenotype and genotype that determine the expression of adhesion molecules and receptors on these cells, stimulating their proliferation and new angiogenesis [7,9].

Here, we discuss the roles of cytokines and growth factors in pathogenetic pathways in MM and as prognostic and predictive biomarkers. We also elaborate on the major genetic and epigenetic aberrations that impact critical pathways responsible for the release of inflammatory and angiogenic cytokines [10]. Finally, we discuss the role of the BM microenvironment in drug resistance to anti-MM drugs [11].

## 2. Pro- and Anti-Inflammatory Cytokines Induce Pro- and Anti-Tumor Effects

Among the pro-inflammatory cytokines produced in the BMME, ILs certainly have the most significant influence on the malignant evolution of PCs. These cytokines make the PCs less differentiated and resistant to apoptosis and ultimately contribute to the malignant transformation from MGUS to MM [12].

The different stages of disease progression from MGUS to smoldering/symptomatic-MM are related to a progressive increase in cytokine levels (e.g., IL-1, IL-6, IL-12, IL-15, IL-17, IL-18, IL-22, IL-23, TNF-a and IFN-g) in the serum of MM patients. An eight-gene signature (IL-8, IL-10, IL-17, CCL3, CCL5, VEGFA, EBI3 and NOS2) involved in B lymphocyte inflammation has also been shown to distinguish the different phases of disease progression (MGUS/smoldering/symptomatic-MM) with an 84% accuracy [13].

The IL-1 family is a group of potent proinflammatory cytokines which includes IL-1α and IL-1β, regulated by different inhibitory receptors (e.g., IL-1Rα, IL-1RII) [14]. IL-1 family members have complex, divergent roles in the control of carcinogenesis and tumor progression [15]. IL-1 is produced by different types of cells in the BMME, such as FBs and B lymphocytes, and indirectly promotes the survival of PCs. In detail, low quantities of IL-1 stimulate the abundant production of IL-6, via a prostaglandin E2 (PGE2) loop [14,16]. According to several studies, the production of IL-1β is associated with the clonal evolution from MGUS to MM, revealing a minimal concentration in MGUS PCs [17,18,19,20] (Figure 1).

IL-2 is predominantly secreted by antigen-activated T cells and supports their proliferation and differentiation shortly after T cell receptor (TCR) stimulation [21], although its effects are controversial. In fact, IL-2 promotes the growth of T and NK cells but simultaneously induces the apoptosis of T cells [22] (Figure 1). IL-2 is used to treat immunogenic tumors such as melanoma, as it interferes with T cell-mediated responses. NK cells in MM disease express PD-L1 independently of IL-2, while in healthy subjects this receptor is expressed only in the presence of IL-2 [23]. Furthermore, hypoxia abrogates the cytotoxic activity of NK cells against multiple myeloma cells, but IL-2 can restore this ability. NK cell-based immunotherapy has been uncovered based on the above-mentioned findings [24].

IL-6, a pleiotropic cytokine with a variety of functions, is involved in immune responses, inflammation, hematopoiesis, bone metabolism and embryonic development, but above all it is an inducer of immunoglobulin production. IL-6 is a prototype member of the IL-6 family of cytokines, which is made up of 10 members, including IL-6, IL-11, IL-27, oncostatin M (OSM), leukemia inhibitory factor (LIF), ciliary neurotrophic factor (CNTF), cardiotrophin 1 (CT-1), factor 1 of cardiotropin-like cytokines (CLCF1), IL-35 and IL-39 [25]. IL-6 promotes MM PCs’ growth through several mechanisms involving signaling pathways with both pro- and anti-inflammatory effects [26]. The autocrine secretion of IL-6 is mainly associated with aggressive phenotypes. Frassanito et al. showed that autocrine levels of IL-6 produced by T cells are higher in patients with active disease than in patients in the plateau phase, remission or MGUS due to anticancer T helper 1 (Th1) cell dysfunction [27]. The paracrine mechanism occurs in the BM environment: IL-6 is secreted by BMSCs, monocytes, ECs, macrophages and FBs stimulated by TGF-β, TNF-α, VEGF, and IL-1. TNF-α induces IL-6 secretion in BMSCs activating NF-kB; on the contrary, the inhibition of NF-kB activity by the specific inhibitor of the IkB kinase (IKK) downregulates the secretion of IL-6 by BMSCs, promoting cell growth [26]. After binding to its receptor, IL-6 activates the signal janus kinase/signal transducer pathway (JAK/STAT) and the mitogen-activated GTPase/protein kinase pathway (RAS/MAPKs), inducing MM cell growth, the inhibition of apoptosis and proliferation. Due to the crucial role of Il-6 in MM progression, the blocking of its signaling pathways has been studied as a possible anti-MM target. This aspect was further investigated when the paracrine production of IL-6 was discovered in myeloid precursors [28]. BAFF and APRIL, new members of the TNF family, have been identified in recent years as potent inducers of MM cells. Both activate NF-kB, phosphatidylinositol-3 (PI-3) kinase/AKT and MAPK pathways and induce a strong up-regulation of the anti-apoptotic proteins Mcl-1 and Bcl-2 (Figure 1). The production of APRIL and its ligand BAFF is mediated by IL-6 production and has been recently considered as a new drug target, such as chimeric anti-IL-6 antibody Siltuximab [29,30].

IL-8, a member of the CXC family of chemokines, is produced by lymphocytes, monocytes, ECs, FBs, hepatocytes and keratinocytes. Substantially, it has a pro-tumor function by altering features of the tumor cells through the transition to a mesenchymal phenotype, to a migratory cell state, or by promoting proliferation [31]. Herrero et al. [32] investigated a possible role for IL-8 in osteoclastogenesis in a breast cancer model, in which IL-8 released by breast cancer cells increases osteoclast formation, probably contributing to bone metastasis. In this regard, it has been shown that in MM patients treated with melphalan and bortezomib, there is an increase in the production of IL-8 by BMSC and osteoclasts which favor bone damage [32]. Hence, the inhibition of IL-8 may represent a therapeutic target to counteract the aggressiveness of MM disease.

IL-11 belongs to the IL-6 family cytokines, along with inhibitory factor of leukemia (LIF), oncostatin M (OSM), ciliary neurotrophic factor, cardiotropin-1 and cytokines similar to cardiotrophin. These cytokines regulate immune responses in the acute phase of infections and also during processes such as hematopoiesis, the regeneration of the liver and neurons, embryonic development and fertility [33]. Accumulating evidence supports the notion that these cytokines and their related receptors play an important role in cancer biology and represent potential biomarkers for disease progression [34,35,36]. Giuliani et al. [37] demonstrated that IL-11 is involved in osteocytes and osteoclasts’ formation, inducing MM bone lesions. According to this study, the number of viable osteocytes decreases proportionally to bone invasion due to increased osteoclasts. On a bio-humoral level, this process corresponds to an increase in the production of osteoclastogenic cytokines, such as IL-11, found to be increased in patients with bone damage [37] (Figure 1).

IL-12 concentrations increase in MM patients compared to MGUS [11]. IL-12, mainly produced by mononuclear phagocytes, dendritic cells and neutrophils [32], is highly associated with pro-inflammatory, immunomodulatory and anti-tumor effects, stimulating the production of IFN-γ. The latter has an anti-angiogenic role by inhibiting the production of VEGF and FGF-2, and by stimulating the release of antiangiogenic chemokines [38] (Figure 1). Thalidomide, the progenitor of immunomodulatory drugs, exerts a robust suppression of the IL-12 production by peripheral blood mononuclear cells (PBMCs) [38]. Wang et al. showed that the synergistic effect of the proteasome inhibition with bortezomib and the immune treatment with IL-12 yields a more significant tumour growth inhibition [39]. In view of these broad-spectrum roles in the regulation of immune responses, IL-12 family cytokines are recognized as possible candidates for the modulation of antitumor immunity [40].

IL-27 is a cytokine that belongs to the IL-12 superfamily, which comprises IL-12, IL-23 and IL-35, which are produced by antigen-presenting cells in response to microbial or host immune stimuli and are involved in the regulation of immune responses against infections and tumor development [41]. IL-27 is composed of the EBI3 and p35 subunits and activates both STAT1 and STAT3 through a single IL-27 receptor, consisting of the WSX-1 receptor subunit coupled with the gp130 chain. The activation of STAT1 and STAT3 in T lymphocytes causes an increase in the proliferation of differentiated of CD4+ T helper and early Th1 cells and the suppression of the differentiation of Th2 and Th17 cells. Furthermore, IL-27 is involved in the production of regulatory T cells [42,43]. Several studies demonstrate that IL-27 activity may vary based on the type of target cell but in general appears to be increased in MM patients with an active disease. IL-27 primarily stimulates the expression of programmed death-ligand 1 (PD-L1) in different cell types, including CD4+ and CD8+ T cells, monocytes, dendritic cells, ECs and cancer cells [41]. In particular, as shown by Dondero et al., IL-27 maintains or increases the functions of NK cells, induced by suboptimal IL-15 concentrations, against ECs isolated from BM aspirates of MM patients. NK cells were shown to kill MM ECs and produce IFN-γ. Finally, IL-27 showed an extraordinary ability to increase the regulation of PD-L2 and HLA-I expression on the tumor endothelium, while it did not modify that of PD-L1 and HLA-II [44]. This therefore also reveals the antiangiogenic properties of IL-27 (Figure 1).

IL-22 belongs to the IL-10 superfamily cytokines which regulate the acute phase response, activating the innate immune system, cell migration and differentiation, and gene expression. Different cell types produce IL-22, including activated Th1 lymphocytes and endothelial cells. Through its receptor (IL22R), IL-22 activates JAK1, tyrosine-protein kinase (Tyk)2 and MAPK signaling pathways and therefore promotes cell proliferation and drug resistance in MM via the phosphorylation of STAT3 (Figure 1). Furthermore, the production of IL-22 in patients with active MM correlates with the release of IL-1β depicting the inflammatory component of the disease [45]. In general, studies conducted on this cytokine show that high secretion levels correlate with poor prognoses.

## 3. Angiogenic Cytokines and PCs Clonal Expansion

PCs function as primary inducers of angiogenesis and are crucial for the activation of BMSCs [46].

PCs express receptors such as αvβ3 integrin, which are important for the interaction with SCs, and secrete cytokines, such as VEGF-A, FGF-2, HGF-SF, angiotensin-1 (Ang-1), IGF-1, C-X-C motif chemokine (CXCL)12/SDF-1α and TNF-α.

These cytokines mediate cell growth (IL-6, IGF-1, SDF-1α, VEGF), survival (IL-6, IGF-1), drug resistance (IL-6, IGF-1, VEGF), migration (IGF-1, VEGF, MMP, SDF-1α) and angiogenesis (VEGF) in the BM.

### 3.1. Vascular Endothelial Growth Factor A (VEGF-A)

Vascular endothelial growth factor A (VEGF-A) belongs to a family of six proteins similar in structure and function: VEGF-A, -B, -C, -D, -E (viral factor), PDGF. These proteins exert their angiogenic (VEGF-A, -E/VEGFR-2-neuropilin-1,-2) activity or lymphangiogenic (VEGF-C, -D/VEGFR-2, -3) activity by binding to their respective receptors [7]. During the angiogenic switch, PCs acquire an angiogenic phenotype. This phenotype is the result of the expression of some oncogenes (c-myc, c-fos, c-Jun, ETS-1) that encode angiogenic factors and a shift from CD45-positive to CD45-negative PCs that are producers of VEGF [47]. Solimando et al. [48] demonstrated that a high expression of intercellular adhesion molecules, such as junctional adhesion molecule A (JAM-A), can essentially stimulate MM-associated angiogenesis.

In addition, genetic studies have introduced gene expression profiling 70 (GEP 70) [49], a microarray-based 70-gene classifier that identifies patients with a high risk for short progression-free survival and overall survival. GEP70 includes markers of angiogenesis, such as FABP5, BIRC5, AURKA, ALDOA, YWHAZ and ENO-1, strong mediators of neo-vasculogenesis [10].

VEGF-A is the most pro-angiogenic cytokine in MM: it is a regulator of the cell growth, survival and migration of ECs through VEGFR-2; in contrast, when acting through VEGFR-1, it regulates the growth of BMSCs.

Tumour-associated hypoxia promotes the production of VEGF-A via hypoxia inducible factor 1α (Hif-1α), an important transcription factor that regulates angiogenesis, mainly through the induction of VEGF transcription [50].

NF-κB, a transcription factor that plays a key role in the survival and proliferation of MM PCs, induces the overexpression of VEGF-A [5,51]. NF-κB can be activated by the phosphoinositide 3-kinase (PI3K)/Akt signaling pathway, implying that PI3K/Akt activation may play a role in angiogenesis and tumour progression in hematological malignancies [52]. VEGF-A directly stimulates the migration, proliferation, and survival of PCs through the autocrine and paracrine loops VEGF-A/VEGFR-2. Specifically, VEGF-A mediates the resistance to apoptosis via HSP90, which binds to Bcl-2 and Apaf-1, suppressing their apoptotic functions [53].

A recent clinical study involving MM patients (GIMEMA-MM0305 NCT01063179) has shown that high levels of VEGF and FGF-2 were associated with an unfavorable prognosis, supporting the notion that VEGF plays an important role in MM progression and suggesting that angiogenic factors may be used as non-invasive prognostic biomarkers in MM [54].

### 3.2. Tumour Necrosis Factor α (TNFα)

Tumour necrosis factor α (TNFα) is a proinflammatory cytokine produced by monocytes, macrophages, lymphocytes, and NK cells, which exerts its pro-tumoral action mainly on MM cells and BMSCs [55].

Bladè et al. reported that high serum concentrations of TNF-α in MGUS patients correlate with a higher probability of malignant progression when compared to patients with low serum concentrations [56].

By binding to its receptor, TNFR1, TNF induces the expression of pro-survival genes via the trimeric IκB kinase (IKK) complex/NKkB axis [57,58].

### 3.3. Insulin-like Growth Factor-1 (IGF-1)

Insulin-like growth factor-1 (IGF-1), secreted by the BMSC and osteoblasts, induces the growth, survival, and migration of cells by binding to its receptor on MM cells’ IGF-1R, with the subsequent activation of MAPK and PI3K/Akt signaling pathways. The Akt cascade leads to the activation of anti-apoptotic proteins Bcl-XL and Bcl-2 which promote PCs’ survival [59]. Furthermore, IGF-1 stimulates MM PCs to secrete VEGF. Finally, it promotes the proliferation, growth and chemotaxis of MMECs and BMSCs [7,59].

### 3.4. Matrix Metalloproteinases (MMP-2, MMP-9, uPA)

Matrix metalloproteinases (MMP-2, MMP-9, uPA) and their inhibitors (TIMP-1 and TIMP-2) are constantly produced by MM PCs (MMP-2 and -9) and BMSC (MMP-1 and -2). MMPs degrade collagen and fibronectin, allowing MM PCs to invade the stroma and the subendothelial basement membrane [60].

### 3.5. Fibroblast Growth Factor-2 (FGF-2)

Fibroblast growth factor-2 (FGF-2) promotes the proliferation, growth, and chemotaxis of MM ECs and BMSCs. It promotes the secretion of IL-6 and VEGF by BM SCs and is found to be upregulated in active MM. Its expression is related to the microvascular density of the BM [7].

FGF-2, together with VEGF and IGF, all secreted by MM plasma cells and inflammatory cells, promotes the recruitment of bone marrow stem cells and progenitor cells into the tumor microenvironment. Thus, MM endothelial cells are activated, participating in the formation of new vessel walls [61,62].

High levels of growth factors and angiogenic cytokines, such as VEGF-A, FGF-2, TNF-α, urokinase and MMPs, are secreted by the precursors of “cancer associated fibro-blasts” (CAFs). CAFs derive from cells that undergo the endothelial–mesenchymal phase or the mesenchymal transition and their precursors are resident FBs and progenitor cells that promote the formation of neovessels [6].

VEGF, FGF-2 and HGF can recruit and activate tumour-associated macrophages. Rajkumar et al. demonstrated that a higher proportion of CD68+ macrophages is present in patients with active MM compared to patients with an asymptomatic disease or MGUS. When exposed to VEGF and FGF-2, BM macrophages are capable of vasculogenic mimicry in vitro, making these cells endothelial-like cells. The latter can generate capillary networks in vitro. Indeed, tumor-associated macrophages express markers of both macrophages and endothelial cells with confocal laser microscopy [63].

### 3.6. Angiopoietin-1 and -2 (Ang-1 and -2)

Angiopoietin-1 and -2 (Ang-1 and -2) expression in MM patient serum and BM samples correlates with the BM microvascular density [64,65,66,67]. Several studies have shown that Ang-1 and Ang-2 are overexpressed in MM cell lines and primary PCs obtained from MM patients [65,68], and that the Tie-2 angiopoietin receptor is upregulated in the BM ECs in the presence of MM cells [69]. High Ang-1 and -2 expression levels were detected in patients with MM versus controls [64] and have been described as independent prognostic factors in these patients.

### 3.7. Platelet-Derived Growth Factor-BB (PDGF-BB)

Platelet-derived growth factor-BB (PDGF-BB) is strongly associated with the VEGF pathway and its altered secretion as well as other proangiogenic molecules, such as VEGF, FGF, EGF, TGFβ, angiopoietin in the tumor microenvironment, contributes to the mobilization and differentiation of EPCs and ECs’ proliferation [70].

Platelet-derived growth factor receptors-PDGFR AB and its α and β receptors in the serum of MM patients show a strong positive correlation with angiogenesis and bone marrow microvessel density. This also results in a prognostic value in terms of overall survival, as a higher PDGFR is associated with significantly higher microvessel density-MVD, which correlates with a lower survival rate [71].

PDGF and the PDGF-BB/PDGF-Rβ kinase axis expressed in the PCs of MM patients promotes tumor progression by activating ERK-1/2 and AKT [71,72]. In addition, PDGF secreted by MM PC and other SCs intervenes in the recruitment and differentiation of monocytes into active macrophages by activating the VEGF-A/VEGFR-1 and FGF-2/FGFR-1,-2,-3 pathways [73,74,75].

### 3.8. Hepatocyte Growth Factor (HGF)

Hepatocyte growth factor (HGF) is a potent angiogenic cytokine that induces the proliferation and migration of ECs by activating their specific tyrosine kinase receptor mesenchymal–epithelial transition factor cMET [74].

The HGF/cMET pathway triggers several signaling pathways involved in tumor growth, angiogenesis and metastatic spread. The HGF/cMET pathway acts on the pathogenesis of MM, enhancing the expression of VEGF/VEGFR-2 in MM ECs. When VEGF and HGF bind to their respective receptors, the dimerization and autophosphorylation of VEGFR and cMet induce the recruitment of signaling proteins to the binding site. Consequently, downstream pathways such as PI3K/AKT and Ras/ERK are activated, generating biological responses of cell survival, angiogenesis and tumor progression [7,74].

## 4. Cytokines and Bone Resorption

Osteolytic bone destruction is the most important effect of the dramatic changes in the BM microenvironment caused by the cross-talk between MM cells and the BM niche [75].

These lesions (“punched”) result from the increase in the number and activity of osteoclasts and the reduction in the number of osteoblasts and, therefore, the poor osteogenesis in the vicinity of MM cells [76].

The balance between the receptor activator of NF-kB-*RANK* and osteoprotegerin-OPG, which has a critical role in the regulation of osteoclastogenesis and bone remodeling activities, is maintained by the activity of different cytokines and pathways (IL-1, IL-6, TNF alpha, TNF/TRAF, PI3K, c-Src, Akt/PKB and mTOR) [77].

Several cytokines and growth factors are involved in the downregulation of osteoblastic activities. Exceptionally fundamental in bone formation and remodeling are the Wnt/dickkopf homolog 1 (DKK1) pathway, IL-3, IL-7, secreted frizzled-related protein-2 (sFRP-2), runt-related transcription factor 2 (Runx2), HGF, and TGF-β [78]. The activation of Runx2 and the Wnt pathways induce the differentiation of resident macrophages in osteoclasts and the transdifferentiation of PCs to functional osteoclasts [78,79].

DKK1 is overexpressed in patients with MM lytic bone lesions. Moreover, it also upregulates the production of Wnt-regulated OPG and RANKL by osteoblasts. If DKK1 is inhibited and Wnt signaling is activated, bone disease in MM is inhibited and the tumour mass is reduced [79]. The number of new, activated osteoclasts increases in areas near PCs, suggesting that bone lesions derive from the local production of osteoclast activation factors (OAF) secreted by MM PCs or SCs [79]. Many osteoclast-activating factors (IL-6, IL-1β, HGF and TNF-α) are released by the mutual interactions of tumour and bone marrow cells with a major involvement of RANKL, the decoy receptor OPG, and MIP-1α a.

RANKL is a member of the TNF family that binds to RANK on osteoclast precursors and promotes the production and differentiation of osteoclasts. OPG is a decoy receptor for RANKL. In MM, RANKL is increased, and OPG is decreased [75,80]. The binding between PCs and BMSCs through vascular cell adhesion molecule 1 (VCAM1) induces the overexpression of RANKL in both cell types, and inhibits the production of OPG by SCs [61,81,82]. MIP-1α, produced by MM PCs, promotes the proliferation and differentiation of osteoclasts’ maturation [83] (Figure 2).

In MM bone disease, osteoclasts can originate from resident monocyte-macrophages stimulated by RANKL overexpressed by the stromal cells [84,85].

It has been shown that CD38 is expressed by effectors and inhibitory cells and by osteoblasts. CD38 also appears to be involved in the formation and differentiation of osteoclasts and may play a role in bone remodeling [86,87]. Hence, there is a rationale for using the anti-CD38 monoclonal antibody (Daratumumab) against MM bone disease and the protection of MM PCs by BMSCs [86,87,88]. Large extracellular vesicles (i.e., microvesicles [4]) released after treatment with Daratumumab have been shown to be enriched in CD38. These microvesicles can be transported through the bloodstream and exert their action at distant sites where they contribute to ADO production [89], immunosuppression and bone remodeling within the BMME, via the modulation of pro- and anti-inflammatory cytokine release. Thus, microvesicles may be considered factors that determine the drug resistance to Daratumumab abrogating anti-MM immune responses [87,88,89].

## 5. Roles of Cytokines in Drug Resistance in Multiple Myeloma

Although the number of anti-MM treatments has dramatically increased in recent years, resulting in higher overall survival (OS) rates (up to 6–10 years), the emergence of drug resistance remains a major challenge for the management of MM [90,91,92]. Accumulating evidence suggests that targeting both PCs and the tumour microenvironment as well as the interactions between (cellular and non-cellular) BMME components and MM PCs may represent the ideal approach for treating MM patients [93]. A summary of anti-MM drugs is provided in Table 1 in which the mechanisms of action are indicated as well as possible implications of cytokines in drug actions or drug resistance.

Resistance to chemotherapy can be acquired or de novo [100]. The latter is innate and therefore occurs prior to drug exposure. Acquired resistance develops following genetic and epigenetic changes that make the tumor cell phenotype more and more complex and resistant to drugs [82]. The acquisition of drug resistance develops when malignant cells become genetically unstable and synthesize excessive quantities of abnormal proteins. Such genetic abnormalities, particularly epigenetic aberrations, influence DNA methylation patterns and histone modifications of genes, especially of tumor suppressor genes [101]. Furthermore, according to recent studies, factors such as microRNAs are involved in these mechanisms, modulating cellular signaling pathways that interfere with cell growth, proliferation, metastasis and drug resistance [102].

BMSCs have been shown to contribute to drug resistance in MM [103,104,105,106,107,108,109]. Different groups have shown that close interactions between MM and BMSC cells, hematopoietic niche components, endothelial cells and the extracellular matrix lead to so-called cell-mediated drug resistance (CAM-DR) [103,104]. CAM-DR is a mechanism in which MM cells escape the cytotoxic effects of anticancer therapy through adhesive interactions with BMSCs and/or ECM components using integrin family adhesion molecules (i.e., CD138/syndecan-1, Vascular Cell Adhesion Molecule-1/VCAM-1, Very Late Activation Antigen/VLA4/α4β1, VLA-5/α5β1, αvβ3 and β7 integrins) [105]. BMSCs are able to control the expression of anti-apoptotic proteins of the bcl-2 family and the ABC drug transporter proteins in myeloma cells, both involved in CAM-DR. Furthermore, BMSCs release IL-6 and IGF-1, which activate the signal transduction pathways that mediate drug resistance and increase the activation of HIF-1, affecting tumor metabolism and drug resistance [95,106,107,108,109,110,111,112,113,114].

*Role of proteasome inhibitors.* MM cells have been shown to be resistant to several drugs, such as doxorubicin, melphalan, vincristine, dexamethasone and mitoxantrone via the above-mentioned CAM-DR mechanisms [95,110,111,112,113,114]. Instead, the proteasome inhibitor bortezomib outperforms CAM-DR to vincristine and dexamethasone by inhibiting the adhesion of MM cells to fibronectin and BMSCs. Conversely, other integrins involved in CAM-DR, such as VLA-5 and β7, increase cell adhesion, migration and MM cell homing to the BM, and reduce bortezomib- and melphalan-induced apoptosis [115,116]. Proteasome inhibitors (i.e., bortezomib, carfilzomib, ixazomib) target the complex action of the proteasome, and by modulating NF-kB, they interfere with critical cellular processes, such as growth arrest, apoptosis, cell cycle progression, inflammation and immune surveillance [117]. The blockade of NF-kB inhibits the adhesion of MM cells to BMSCs, reduces the secretion of VEGF-2 and FGF-2 and angiogenic cytokines (e.g., VEGF, IL-6, IGF-I, Ang-1 and Ang-2), and consequently stops the growth of MM cells and creates new, activated ECs [118].

In addition to cell adhesion, numerous other mechanisms underlie the onset of primary and secondary resistance. The overexpression of the proteasome β5 subunit was the most frequent variation observed in bortezomib-resistant cell lines. The increase in the expression of the transcription factor of the oncogene c-Maf also affects the resistance to therapy with bortezomib and carfilzomib [108]. Finally, Zheng et al. [94] observed that the interaction between cytokine–cytokine receptors, autophagy, the ErbB signaling pathway and microRNAs can play a role in the resistance to carfilzomib in MM.

*Role of Immunomodulatory drugs (IMIDs).* IMIDs (thalidomide, lenalidomide, pomalidomide) selectively enhance the degradation of the transcription factors Ikaros (IKZF1) and Aiolos (IKZF3) vital for B-cell growth by binding to cereblon. In addition to their tumoricidal activity, IMIDs have critical antiangiogenic properties by reducing the expression of angiogenic cytokines [119,120]. They modulate the TNF-α signaling via direct and indirect effects on the tumor microenvironment [121], and they reduce the secretion of FGF-2 [112], VEGF and IL-6 by both MM cells and BMSCs [59].

Cell adhesion promotes the release of cytokines, chemokines and other factors that promote angiogenesis and immune suppression [10,59]. After extravasating into new tissue microenvironments, accommodating to survival signals, and colonizing extramedullary ecosystems, few plasma cells survive in the circulation [122,123].

Mechanisms of resistance to IMIDs studied to date include the downregulation of cerebron expression due to mutations in its gene. A recent study by Kortum et al. [96] identified three cerebron mutations developed by clonal selection after prolonged IMiD therapy.

New generations of more powerful cereblon E3 ligase modulators (CELMoDs^®^)—iberdomide (IBER) and CC-92480, are currently under investigation [99]. Their mechanism of action involves enhanced interactions with cereblon or substrates due to more extensive structures containing additional phenyl and morpholino fractions [124].

*Role of immunotherapy.* Increasing insights into the biology of MM, including the associated immune dysregulation, and the development of several immune-based therapies have renewed scientific interest in the clinical application of immunotherapy in MM. Monoclonal antibodies (mAbs) exert their cytotoxic function through different mechanisms: antibody-dependent cellular cytotoxicity (ADCC) through the engagement of immune effector cells, complement activation, antibody-dependent phagocytosis, and direct effects on target cells acting through different signaling pathways. Several mAbs are currently approved for the treatment of MM: elotuzumab, an IgG1κ mAb with a specificity against SLAMF7, daratumumab and isatuximab, humanizeds IgG1 mAb that targets CD38 [125,126]. Resistance to daratumumab is probably associated with an alteration in the CDC mechanism mediated by the overexpression of the complement inhibitory proteins CD55 and CD59 [97,126].

*Role of B cell maturation antigen (BCMA) and CAR-T cells.* Recently, BCMA has become an essential target for developing novel immunotherapeutics in MM. Targeted therapies against BCMA under evaluation include antibody-drug conjugates (ADCs), bispecific T cell engineers (BITE), and antigen receptor-modified T cell (CAR) chimeric therapies [127,128].

CAR T cells are genetically modified T cells with chimeric antigen receptors expressing a CAR against specific tumor-associated antigens. Binding to the specific antigen activates T lymphocytes, leading to cell lysis and death [129]. The immunosuppressive landscape of tumours is a critical obstacle to the effector activity of CAR-T cells. The tumor microenvironment, including its non-cellular components, various immune cells, abnormal tumor vasculature, immunosuppressive molecules and tumor metabolites, prevents CAR-T cells from exerting a high cytotoxicity. For example, fibroblasts and mesenchymal cells form physical barriers against the entry of T cells. Furthermore, the migration of T cells towards tumor lesions is increasingly challenged by the dysregulation of adhesion molecules, the mismatching of tumor-derived chemokines, and chemokine receptors expressed by immune cells.

Some immunosuppressive cytokine and chemokine as IL-2 recruit Treg cells, MDSCs and tumour-associated macrophages (TAMs) to specifically suppress cytotoxic T cell function. MDSCs have been shown to have a detrimental effect on CAR T cells because of their potent immunosuppressive capabilities directly targeting effector T cells [98].

Several studies have endeavored to integrate chemokines and CAR-T cells to fight cancer, with ongoing investigations focused on CXCR2-modified CAR-T cells that have been shown to induce complete tumour regression and immunological memory in aggressive tumors [130].

Furthermore, as demonstrated by several clinical studies as KarMMa and CARTITUDE-1, CAR- T can specifically recognize BCMA and kill BCMA-expressing MM cells [131].

Recent investigations have also shown that BCMA may represent a biomarker for the diagnosis of, monitoring of and therapeutic response to MM treatments, in both MM patients with secretory features (i.e., presence of tumour-related factors in peripheral blood) and in patients with non-secretory MM, in which it is more difficult to track the disease using conventional markers. Therefore, circulating BCMA may be helpful in detecting MM and also drug resistance [132].

As most of the therapies mentioned in this paragraph target the BMME and restore/activate an anti-MM immune response, BMME is believed to play a key role in therapy resistance for these drugs. As discussed by Neumeister et al. [133], BMME targeting drug resistance may be caused by the cross-talk of BMME and MM cells via soluble factors (e.g., IL-6, APRIL, growth factors), and via the integrin-mediated cell adhesion and Notch signaling, resulting in the inhibition of apoptosis [134,135,136], on which the major cytotoxic machinery of immune cells relies [137,138,139,140]. Drug resistance may also arise following the recruitment of immunosuppressive cells, including MDSCs, Tregs, Bregs and TAM, into the BMME of MM. These immunosuppressive cells might secrete nitric oxide (NO), arginase, reactive oxygen species (ROS), prostaglandin E2 (PGE2), or indoleamine 2,3-dioxygenase (IDO) [141,142,143] as well as immunosuppressive cytokines (e.g., IL-10 and TGF-β), inhibiting the proliferation and expansion of Th1 cells, CTLs and NK cells [144,145]. Furthermore, it has been shown that MSCs exert immunosuppressive actions mediated by the secretion of factors, such as IL-6, TGF-β, IL-10, PGE2, and by the upregulation of surface proteins, such as VCAM-1, ICAM-1, and CD40 [134,142,146,147,148,149]. For therapeutic approaches employing CAR T cells as well as bispecific antibodies, it was observed that MSCs could protect MM cells from highly lytic BCMA-CAR T cells [150], whereas the lytic function of BCMA/CD3 bispecific antibodies was not that much influenced by MSCs [151].

## 6. Conclusions

Cytokines play an important role in the pathogenesis and progression of MM. Some cytokines are produced directly by neoplastic cells as an effect of genetic mutations due to clonal evolution [152], and proinflammatory cytokines appear to be reduced in patients with MGUS compared to those with MM while anti-inflammatory cytokines can be both increased and reduced in MGUS. As described in this review, the role of cytokines in cancer and inflammation can be ambivalent. For this reason, the most critical proinflammatory cytokines involved in tumor progression, such as IL-6 and IL-1, cannot represent therapeutic targets without compromising the patients’ antitumor immunocompetence [153]. Anakinra, the recombinant form of IL-1Ra used for rheumatoid arthritis and periodic syndromes associated with crypirin [154], may be used in specific studies to evaluate IL-1 inhibition in MM.

The inhibition of proangiogenic cytokines has also been widely used in MM therapy. In addition to proteasome inhibitors and immunomodulators that have “indirect” antiangiogenic activity, bisphosphonates are other compounds that, although initially used to reduce bone loss in MM due to their anti-osteoclastic activity, have also been shown to have antiangiogenic activity [155,156]. Zoledronic acid has direct cytotoxic activity on cancer cells, suppresses angiogenesis, inhibits the FGF-2- and VEGF-dependent proliferation of endothelial cells, and inhibits VEGFR-2 in an autocrine cycle [157].

Future studies should aim at better understanding the delicate balance between the various cytokines in MM. The use of nanotechnology has been proposed to solve some of the limitations of conventional drug delivery systems, such as non-specific biodistribution and off-target effects, and to use cytokines for therapeutic purposes [158].

These strategies should aim at targeting both the malignant compartment (PCs) and the tumour microenvironment to overcome the onset of drug resistance and to possibly cure patients with MM.

## Figures and Tables

**Figure 1 jcm-11-06491-f001:**
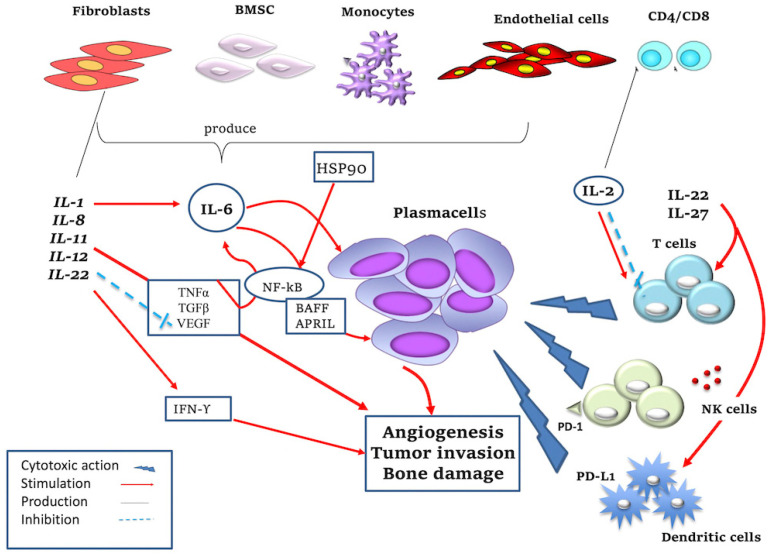
Mechanism of action of pro- and anti-inflammatory cytokines. IL-1, produced by various cell types in the BM microenvironment (i.e., fibroblasts, B lymphocytes), promotes PCs’ survival by stimulating IL-6 production. IL-2 is produced by CD8^+^ and CD4^+^ T lymphocytes and promotes the growth of T and NK cells but at the same time induces apoptosis of T cells; it also stimulates the cytotoxic action of NK cells against malignant PCs. IL-6 promotes the growth of MM PCs and resistance to apoptosis; IL-6 is secreted by BMSCs, monocytes, endothelial cells, macrophages and fibroblasts thanks to the action of TGF-β, TNF-α, VEGF, IL-1. IL-1 and TNF are potent inducers of IL-6 synthesis. In detail, they regulate IL-6 gene expression at the level of transcription, which is mediated by upregulation of NF-kB activation. IL-6 stimulates the production of APRIL and BAFF, which activate NF-kB, phosphatidylinositol-3 (PI-3) kinase/AKT and MAPK pathways, promoting MM cells’ survival. IL-8 has a pro-tumor function promoting angiogenesis and tumor invasion. IL-11 promotes tumor invasion and in particular bone damage. IL-12 stimulates the production of IFN-γ with both pro-inflammatory and anti-angiogenic roles and this inhibits the production of VEGF. IL-22 is produced by T lymphocytes, activates the innate immune system and also regulates cell proliferation and resistance to drug-induced cell death in MM cells. IL-27 has antitumor activity, causing increased proliferation and differentiation of T cells. IL-27 stimulates the expression of immune checkpoint inhibitor programmed death (PD)-1 or PD-L1 in different cell types (i.e., lymphocytes and dendritic cells) and increases NK cells’ activity. HSP and HSP90 are produced by macrophages/monocytes and promote MM cell survival through TNF-mediated and NF-kB signaling pathways.

**Figure 2 jcm-11-06491-f002:**
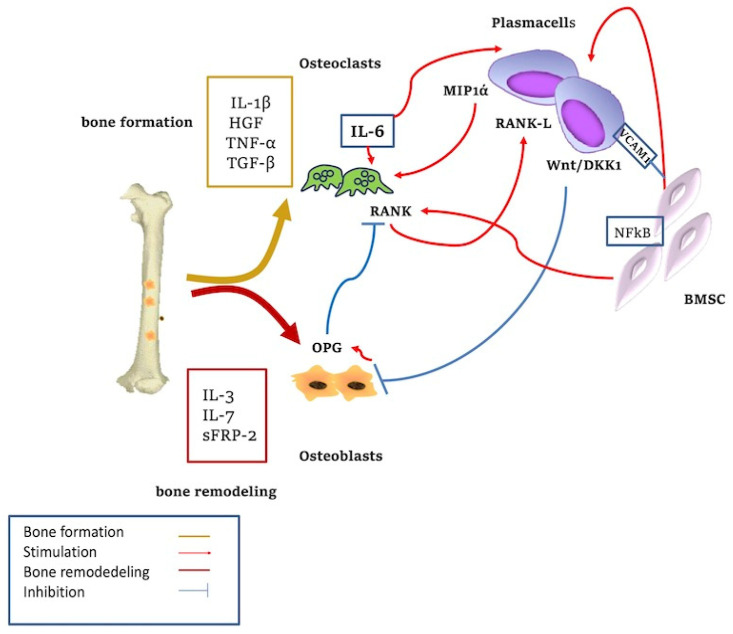
Cytokine-mediated bone remodeling. The adhesion of plasma cells to stromal cells increases the regulation of IL6 with the activation of osteoclasts. The Wnt/DKK1 pathway is overexpressed in MM-associated lytic lesions. IL-3, IL-7, sFRP-2, and Runx2 are involved in bone formation; HGF and TGF-β act in bone remodeling. Wnt signaling stimulates osteoblasts’ (OB) differentiation. Blockade of this pathway through DKK1 inhibits OB formation. Wnt signaling upregulates RANKL expression from OB precursors, resulting in increased osteoclastic activity and bone resorption. Activation of Wnt pathway increases OPG production from OB which in turn downregulates RANKL-driven osteoclastogenesis. MIP-1α, produced by MM PC, promotes the proliferation and differentiation of osteoclast maturation. MM cells inhibit the osteoblastic activity through inhibitory factors and reduced production of cytokines.

**Table 1 jcm-11-06491-t001:** Drugs, mechanisms of action, involved cytokines and related clinical studies.

*Drugs*	*Mechanism of Action*	*Cytokines Involved*	*Resistance*	*References*
Proteasome inhibitors (Bortezomib, carfilzomib ixazomib)	Blockade of NF-kB; interferes with growth arrest, apoptosis, cell cycle progression, inflammation and immune surveillance	Inhibits adhesion of MM cells to BMSCs; reduced production of VEGF-2, FGF-2,IL-6, IGF-I, Ang-1 and Ang-2	Interaction between cytokine-cytokine receptors; autophagy	Zheng et al., 2017 [94] Furukawa Y et al., 2016 [95]
IMIDs(Thalidomide, Lenalidomide pomalidomide)	Selectively enhance degradation of the transcription factors IKZF1 and IKFZ3; anti-angiogenic properties	Modulate TNF-α;reduction of FGF-2,VEGF, IL-6 by MM cells and BMSCs;reduce adhesion between MM PCs and BMSC	Downregulation of cerebron expression	Kortum et al., 2016 [96]
Monoclonal antibodies (elotuzumab, daratumumab, isatuximab)	Antibody-dependent cellular cytotoxicity (ADCC), complement activation, antibody-dependent phagocytosis, direct effect on target cells	Effects mediated by INFγ and TNFα	CDC mechanism mediated by the overexpression of CD55 and CD59	Bellone et al., 2012 [97]
Anti-BCMA BITE CAR	ADC inducing cytotoxic cell death Simultaneous binding to T cell and tumor antigens Chimeric antigen receptors against specific tumor-associated antigen	IL-6, IL-10, TNFα inducing a (CRS)	immunosuppressive TME, production IL-2 by Treg MSC protection	Kankeu Fonkoua et al., 2022 [98]
CELmoDs	rapid degradation of distinct cell types mediated by physical interactions with the CRBN/DDB1 complex	IL-2 up-regulation		Thakurta et al., 2021 [99]

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
