# Peer review of "Pathways of Angiogenic and Inflammatory Cytokines in Multiple Myeloma: Role in Plasma Cell Clonal Expansion and Drug Resistance"

_jcm, 2022, doi:10.3390/jcm11216491_

Round 1

Reviewer 1 Report (Previous Reviewer 2)

The manuscript “Pathways of angiogenic and inflammatory cytokines in multiple myeloma: role in plasma cell clonal expansion and drug resistance” represents a comprehensive literature review on the role of cytokines in multiple myeloma. Authors have discussed the pathogenetic role of multiple cytokines in a well-structured way, clearly showing the association with innovative treatment options. Thus, the topic is clinically relevant and important. 

Authors have responded to previous comments with extensive modifications throughout the manuscript therefore I recommend to accept the article for publication in its current version. I have no further critical remarks. 

Author Response

thanks for your comment

Reviewer 2 Report (Previous Reviewer 3)

The authors systematically revealed the current evidence focusing on angiogenic / inflammatory cytokines and microenvironment in multiple myeloma (MM) development and its drug resistance and the biological target drugs that block MM cell proliferation and survival, MM cell and stromal cell interaction. This paper is useful for the readers of J clin Med since it is concise and has lucid explanations and figures. 

In the revised version, there are depicts of alteration of angiogenic / inflammatory cytokines and microenvironment after novel target therapy, such as anti- BCMA (B-cell maturation antigen) antibody based conjugate or CAR T-cells.

Author Response

thanks for your comment

This manuscript is a resubmission of an earlier submission. The following is a list of the peer review reports and author responses from that submission.

Round 1

Reviewer 1 Report

Major changes needed:

1. Section 2: Pro- and anti-inflammatory cytokines induce pro- and anti-tumor effects: This section needs to be re-organized. In the section 2.1 for IL-1 you discuss other cytokines/interleukins, which are also mentioned thereafter.

2. Below the line 205 – You need to make a new sub-section here. Last sub-section is 2.6. Interleukin-12 (IL-12), however is do not think you still refer to IL-12 from line 205 down.

3. Please read the review carefully, and fix grammatical and languages errors - too many to be mentioned. Please pay extra attention to coherence of the manuscript.

 Minor changes needed:

Line 21: “soluble components stimulation” it sounds grammatically incorrect, if able please rephrase.

Line 22: “MM PCs” – please substitute with either “MM cells” or “malignant PCs”

Line 23: substitute “mediates of” with “mediated by”

Line 25: substitute “forming with “the formation of”

Line 28: remove the word “microenvironment” – BMME is sufficient per aforementioned abbreviation explanation

Line 31: substitute “revise” with discuss

Line 39: substitute “cells” with “cell”

Line 39-40: rephrase: “MM is an incurable plasma cell dyscrasia”, remove “still incurable today”

Line 40: substitute “moves to” with “most commonly starts with”

Line 42: add “years of age” after 25

Line 42: switch 1% to 1-2% (some high risk MGUS have higher risk of yearly progression)

Line 42: substitute “progress in” with “progress to”

Line 46: minor grammatic error: substitute the word “eluding” with “elude”, and “resisting” with “develop resistance to”

Line 50: please substitute with either “MM cells” or “malignant PCs”

Line 64: “BMME tumor microenvironment”, would rephrase as “tumor BMME”

Line 66: substitute “thanks to” with “via the”

Line 69: “these deregulated networks of alterations”, can you please rephrase to be more understandable?

Line 70: “this niche", do you mean vascular niche?

Line 71: “is currently the direction researchers take to understand the MM drug resistance mechanisms, can substitute with something like “is currently under rigorous investigation to better understand the underlying drug resistance mechanisms in MM”

Line 72: substitute “responsible for” with “play a critical role in”

Line 72: can remove “development of”, it is redundant

Line 73: substitute “more generally” with “In general”

Line 73-76: would substitute “the prevalence of proinflammatory cytokine secretion” with “the balance between the secretion of proinflammatory cytokines […………] and anti-inflammatory cytokines ………

Line 78” substitute “cell’s” with “cells”

Line 78: “Consequently, considering MM ECs, we consider…”, can you please rephrase to be more understandable?

Line 82-84: “This review focuses on the genetic and evolutionary heterogeneity in MM based on various genetic and epigenetic aberrations that influence the most critical pathways on which the release of inflammatory and angiogenic cytokines depends” would rephrase as: “This review focuses on the evolutionary heterogeneity of MM, elaborating on the major genetic and epigenetic aberrations that impact critical pathways responsible for the release of inflammatory and angiogenic cytokines”

 Line 84-85: would rephrase: Recent studies have highlighted the role of epigenetic modifications in the dedifferentiation of malignant PCs and development of drug resistance

Line 86-87: this sentence does not make sense. Please fix.

Line 90: remove the word “many”

Line 92: “…..and constitute the "humoral" cause of the progression from MGUS to MM”, rephrase: “....ultimately contributing to the malignant transformation”. These cytokines contribute to the malignant transformation, but they are not the only cause.

Line 96: remove the word “find”

Line 102-103: “….indirectly act by promoting PCs survival of PCs as low quantities of IL-1 stimulating the production of high quantities of IL-6, through IL-6 generation via a prostaglandin E2 (PGE2) loop”, would rephrase: “…..indirectly promotes survival of PCs. In detail, low quantities of IL-1 stimulate abundant production of IL-6, via a prostaglandin E2 (PGE2) loop.”

Line 110-111: can remove the first sentence, this has already been mentioned above.

Line 114: please substitute with either “MM cells” or “malignant PCs”

Line 115-117: This need to be written a bit better: IL-1 and TNF are potent inducers of IL-6 synthesis. In detail they regulate IL-6 gene expression at the level of transcription, which is mediated by upregulation of NF-kB activation.

Lines 121-122: grammar needs to be fixed

Line 124: substitute “but” with “and”

Line 125: substitute “activity; causes”, with :activity, causing”

Line 127: add “immune checkpoint inhibitor” before “….Programmed death (PD)-1 PD-L1”

Line 128: substitute “functions” with “activity”

Line 128: HSP and HSP90 are chaperone proteins, why are they mentioned in the cytokine section?

Line 131: remove the word “more”

Line 132: substitute “intervenes in” with “interferes with”, add the word “cell” after “T”

Line 136: “MM NK cells” there are no such terminology, you can say “NK cells in MM patents”, or “NK cells in MM disease”

Line 136: substitute “Programmed death (PD)-1 PD-L1”with “PD-L1”

Line 138: substitute “killing ability” with “cytotoxic activity”

Line 146: substitute “phase plateau” with “plateau phase”

Line 149: “thanks to the action” please change this to a more formal wording

Line 153: substitute “By binding its receptor” with “The binding to its receptor”

Line 155: substitute cells with cell

Line 158: I think you mean “discovered in” not “discovered by”

Line 159: substitute “survival factors for” with “inducers of”

Line 167: substitute “from” with “by”

Lines 172-176: This sentence does not make sense. Not sure what you mean.

Line 178-179: need to add “along with” before inhibitory factor of leukemia (LIF)

Line 184: substitute “has” with “is”

Line 199: substitute “immunomodulating anti-MM drugs” with Immunomodulatory drugs 

Line 202: substitute “determines” with “yields”

Line 303: remove “depending on the age at diagnosis

Line 335: “…therapy based on dual anti-MM activity and MM PCs than on the micro environment”, this does not make sense. Do you mean “dual anti-MM activity including plasma cell and microenvironment directed therapy?”. Please clarify.

Line 356-368: Please break down to 2 sentences, in order to be more easily read

Line 375: maybe new paragraph here?

Line 384-385: “Their activity appears to be mediated through cereblon, the primary putative teratogenic thalidomide”, would just make it simple saying “IMiDs primary work by binding to the cereblon and selectively enhancing the degradation of the transcription factors Ikaros (IKZF1) and Aiolos (IKZF3), which are vital for B-cell growth.”

Line 385: remove “direct”

Line 395: “original drug”, do you mean thalidomide?

Line 398: substitute “concern” with “include”

Line 402: add “of” after “new generations”

Line 403: substitute “study” with “investigation”

Line 402-405: Break this to 2 sentences.

Line 407-407: “In the last 7 years, a better understanding of the biology of MM and its immune dysfunction and the development of various immune-based therapies have led to renewed interest in immunotherapy's clinical potential for treating this disease”, wring language, try to re-phrase as: “Over the past few years, increasing insight in the biology and accompanying immune dysregulation of MM as well as the development of various immune-based therapies……”

Reviewer 2 Report

The manuscript “Pathways of angiogenic and inflammatory cytokines in multiple myeloma: role in plasma cell clonal expansion and drug resistance” represents a literature review on the pathogenetic role of cytokines in multiple myeloma. Authors have assessed the topic in relation to innovative medications. The review is based on 139 sources (listed as 140 entities, but there is an error in numbering: see further, please). It is comprehensive, well-written and interesting. The article is characterised by clear design, logical structure and reasonably high level of English language, ensuring good scientific comprehensibility.

Few aspects could be clarified in order to further improve the manuscript.

1) Please, specify the aim of the study (lines 31 – 33 and 82 – 88). Currently, the title is concentrated on pathogenesis, while the aim (in Abstract) is defined as a literature review on biological target drugs, and the focus in Introduction is placed on heterogeneity.

2) Unfortunately, only 18% of the references are published within the preceding 5 years, and many are from the previous century. Although old sources definitely can be cited when appropriate, a new literature review should emphasize the most recent findings. Please, increase the impact of new articles, published within the last 5 years.

3) Several innovative drugs are mentioned throughout the article. A summarising table on these medications would be of great value for the readers. It could be advised to list the names and pharmacologic mechanisms of these drugs as well as trial data and references.

4) There are multiple formatting issues that does not decrease the scientific value of the article but still should be corrected (e.g., contact details; width of the text/right border in Abstract, page 3 (line 99 – 107) and page 4 (lines 131 – 135; 141 – 143 and 166 – 176); interrupted line (page 8, line 348; page 13, line 597; page 13, line 629); extra spaces (page 8, line 356; page 10, line 474; page 12, line 559; page 12, line 560) and missing spaces (page 13, line 608; page 15, line 743).

5) Please, format References in accordance with the Instructions for Authors. Bibliographic data are incomplete for several sources, e.g., Ref.5, Ref.6, Ref.8, Ref.11, Ref.91 – please, add the missing information and check the whole list of References for accuracy of bibliographic data.

Reviewer 3 Report

The authors systematically revealed the current evidence focusing on angiogenic / inflammatory cytokines and microenvironment in multiple myeloma (MM) development and its drug resistance and the biological target drugs that block MM cell proliferation and survival, MM cell and stromal cell interaction. This paper is useful for the readers of J Clin Med since it is concise and has lucid explanations and figures, however there is a little depiction of alteration of  angiogenic / inflammatory cytokines and microenvironment after novel target therapy, such as anti- BCMA (B-cell maturation antigen) antibody based conjugate, CAR T-cells, bispecific T-cell engagers etc.